

# Organic and inorganic carbon and their stable isotopes in
# surface sediments of the Yellow River Estuary
**Zhitong Yu[1], Xiujun Wang[1*], Guangxuan Han[2], Xingqi Liu[3], and Enlou Zhang[4]**
[1] College of Global Change and Earth System Science, Beijing Normal University,
Beijing 100875, P.R. China
[2] Yantai Institute of Coastal Zone Research, Chinese Academy of Sciences, Yantai,
Shandong 264003, P.R. China
[3] College of Resource Environment and Tourism, Capital Normal University, Beijing
100048, P.R. China
[4] State Key Laboratory of Lake Science and Environment, Nanjing Institute of
Geography and Limnology, Chinese Academy of Sciences, Nanjing 210008, P.R. China
* Corresponding to: Xiujun Wang (xwang@bnu.edu.cn)
**Abstract**
Estuarine sediment is an important carbon reservoir, and thus may play an important role in the global
carbon cycle. The Yellow River Estuary is a large estuary in northern China, having implications for the
Bohai Sea's carbon cycle. However, little is known about carbon dynamics in the sediment of the
transitional zone near the river mouth. In this study, we collected 15 short sediment cores from the
Yellow River Estuary, and measured grain size, total nitrogen (TN), total organic carbon (TOC) and
inorganic carbon (TIC) and the isotopic compositions of TOC ($\delta^{13}C_{org}$) and carbonate ($\delta^{13}C_{carb}$ and
$\delta^{18}O_{carb}$). We found that TIC concentration (6.3-20.1 g kg$^{-1}$) was much higher than TOC (0.2-4.4 g kg$^{-1}$)
in the surface sediment. Both TOC and TIC were higher to the north (2.6 and 14.5 g kg$^{-1}$) than to the
south (1.6 and 12.2 g kg$^{-1}$), except in the southern bay where TOC and TIC reached 2.7 and 15.4 g kg$^{-1}$,
respectively. The $\delta^{13}C_{org}$ value ranged narrowly from -24.26‰ to -22.66‰, indicating that TOC might



be mainly autochthonous. However, C:N ratio varied from 2.1 to 10.1, with higher ratio found in the
southern bay. We estimated that 60.8% of TOC might be from terrigenous OC in the southern bay. The
lower TOC values in the south section were due to relatively higher kinetic energy level whereas the
higher values in the bay was attributable to terrigenous matters accumulation and lower kinetic energy
level. There was a significantly positive correlation between TIC and TOC, indicating that TIC was
primarily from autogenic carbonate. However, the southern bay revealed the most negative $\delta^{13}C_{org}$ and
$\delta^{13}C_{carb}$, suggesting that there might exist some transfer of OC to IC in the section. Our study points out
that the dynamics of sedimentary carbon in the Yellow River Estuary is influenced by multiple and
complex processes, and highlights the importance of carbonate in carbon sequestration.

## 1  Introduction

The rate of $CO_2$ build-up in the atmosphere depends on the rate of fossil fuel combustion
and the rate of $CO_2$ uptake by the ocean and terrestrial biota. About half of the
anthropogenic $CO_2$ has been absorbed by land and ocean. Large rivers that connect the
land and ocean may play an important role in the global carbon cycle (Bianchi and Allison,
2009;Ran et al., 2015;Wang et al., 2016c). On the one hand, river can transport a
significant amount of dissolved and particulate carbon materials from the land to the ocean,
which are subject to recycling and sedimentation in the estuaries, or further transportation
to the marginal seas (Cole et al., 2007;Bauer et al., 2013). On the other hand, there may be
high levels of nutrients in the river waters, which could cause enhanced biological uptake
of $CO_2$ and subsequent carbon burial in the estuaries (Cai, 2010;Raimonet and Cloern,

48  2017).

The Yellow River, the second largest river in China following the Yangtze River,
provides approximately 50% of the freshwater discharged into the Bohai Sea every year
(Wang et al., 2006). However, as the world's largest carrier of fluvial sediment, its
sediment load has continually decreased since the 1950s due to changes in water discharge
and sediment concentration by anthropogenic changes (Wang et al., 2016a). These



changes may have profound impacts on the physical, biogeochemical and biological
processes in the Yellow River Estuary.
There were some studies on sedimentary organic carbon around the Yellow River
Estuary, which were mainly conducted in the Yellow River Delta (Bianchi and Allison,
2009;Ye et al., 2015;Zhao et al., 2015) and in the shelf sediments of the Bohai Sea (Hu et
al., 2016;Liu et al., 2015;Xing et al., 2016;Wang et al., 2017). Limited studies showed a
large spatial variability in total organic carbon (TOC, 0.7-7.7 g kg$^{-1}$) in the Yellow River
Estuary (Li et al., 2014b), with the highest contribution (40-50%) of terrestrial organic
carbon found near the delta, which might be due to the hydrodynamics constrained
sedimentary environment and deposition rate and current speed (Liu et al., 2015).
However, little is known about the TOC dynamics in the sediment for the transitional zone
near the river mouth.
There were limited studies of inorganic carbon dynamics in the Yellow River Estuary.
A field based analysis demonstrated that rate of $CaCO_3$ precipitation was modestly higher
than biological production in the water columns of the estuary (Liu et al., 2014). In
addition, Gu et al. (2009) found that particulate inorganic carbon (1.8% ±0.2%) was also
much higher than particulate organic carbon (0.5% ±0.05%) in the Yellow River Estuary.
These findings indicate that there might be much more inorganic carbon (TIC) than TOC
in the sediment of the Yellow River Estuary. While there was evidence of high level of
TIC in the sediment of the lower Yellow River Delta (Zhao et al., 2015), little has been
done to evaluate the magnitude and variability of TIC in the Yellow River Estuary.
Recent studies have showed that there was a large amount of carbonate in the soils of
lower part of the Yellow River Basin, and higher level of carbonate was associated with
high level of organic carbon (Guo et al., 2016;Shi et al., 2017). One may expect a similar
phenomenon in the sediment of the Yellow River Estuary. The objectives of this study
were to investigate the magnitudes and spatial distributions of TOC and TIC in the surface
sediment of the transitional zone near the river mouth, to evaluate the relationship between
TOC and TIC, and explore the underlying mechanisms that regulate the carbon burial in
the Yellow River Estuary.






## 2   Materials and Methods

### 2.1   Site description

The Yellow River Estuary is a typical river-dominated estuary with weak tides, showing a

tidally affected zone of approximately 10-20 km upriver (Figure 1). The Yellow River

Delta has a warm-temperate continental monsoon climate with distinct seasons. The

annual mean air temperature and rainfall are 11.5-12.4 ℃ and 530-630 mm, respectively;

approximately 70% of the total annual precipitation occurs in the summer, and the pan

evaporation exceeds 1500 mm (Kong et al., 2015;Gao et al., 2016). In the Yellow River

Estuary, monthly water temperature is 4.1 ℃ in January and 26.7 ℃ in July, and annual

wind speed ranges from 3.1 to 4.6 m s$^{-1}$ in the estuary (Shen et al., 2015). The estuary is

characterized by a high sediment load (mainly composed of silt) in the water column,

produced largely by the erosion from the China's Loess Plateau. Most of the sediments

discharged from the modern Yellow River mouth are trapped in the subaqueous delta or

within 30 km of the delta front by gravity-driven underflow (Zhao et al., 2015;Kong et al.,

2015). In recent decades, the annual water and sediment fluxes have declined dramatically,

which is caused by reginal climate change, reservoir construction, and irrigation-related

withdrawals (Shen et al., 2015;Liu et al., 2014).

### 2.2   Field sampling and analyses

During October 2016, we collected 15 short sediment cores (H series) from the Yellow

River Estuary using a Kajak gravity corer and 10 surface soil samples at 7 sites (S1-S7)

along its upstream wetland (Figure 1b). Each sediment core was carefully extruded and cut

into 1-cm interval, and then placed in polyethylene bags which were kept on ice in a

cooler during transport. In the laboratory, we took the top 2 cm sediment and 0-5cm soil

sample, and then freeze-dried for 48 h before analyses.

Grain size was determined using a Malvern Mastersizer 2000 laser grain size

analyzer. According to Yu et al. (2015), each sediment sample and soil sample (~0.5 g)



was pretreated, in a water bath (at 60-80 ℃), with 10-20 ml of 30% $H_2O_2$ to remove
organic matter, and with 10-15 ml of 10% HCl to remove carbonates. The pretreated
samples were then mixed with 2000 ml of deionized water, and centrifuged after 24 hours
of standing. The solids were dispersed with 10 ml of 0.05 M $(NaPO_3)_6$, and then analyzed
for grain size (between 0.02 and 2000 μm). The Malvern Mastersizer 2000 automatically
outputs the median diameter d(0.5) (μm), the diameter at the 50th percentile of the
distribution, and the percentages of clay (< 2 μm), silt (2-64 μm) and sand (> 64 μm)
fractions.

118         C and N contents were measured using an Elemental Analyzer 3000 (Euro Vector,

Italy) at the State Key Laboratory of Lake Science and Environment, Nanjing Institute of
Geography and Limnology, Chinese Academy of Sciences. Freeze-dried samples were
ground into a fine powder, then placed in tin capsules, weighed and packed carefully. For
the analysis of TOC, a ~0.3 g sample was pretreated with 5-10 ml 2M HCl for 24h at room
temperature, and then dried overnight at 40-50 ℃ to remove carbonate. TC and TN were
analyzed without pretreatment of HCl, and TIC was calculated as the difference between
TC and TOC.

126         For the analyses of $^{13}C$ in TOC ($\delta^{13}C_{org}$), approximately 0.2 g of the freeze-dried

sample was pretreated with 5-10 ml 2M HCl for 24 h at room temperature to remove
carbonate, and then mixed with deionized water to bring the pH to 7, and dried at 40-50 ℃
before analyses. Each pre-treated sample was combusted in a Thermo elemental analyzer
integrated with an isotope ratio mass spectrometer (Delta Plus XP, Thermo Finnigan MAT,
Germany). Additionally, $^{13}C$ and $^{18}O$ in carbonate ($\delta^{13}C_{carb}$ and $\delta^{18}O_{carb}$) were measured
following reaction with 100% phosphoric acid on a stable isotope ratio mass spectrometer
(Thermo-Fisher MAT 253, Germany), at the Nanjing Institute of Geology and
Paleontology, Chinese Academy of Sciences. All the isotope data were reported in the
conventional delta notation relative to the Vienna Pee Dee Belemnite (VPDB). Analytical
precision was 0.1‰ for $\delta^{13}C_{org}$ and $\delta^{13}C_{carb}$, and 0.2‰ for $\delta^{18}O_{carb}$.





## 2.3 Statistical methods and mapping
Correlation analyses were performed using the SPSS Statistics 19 for Windows. Spatial
distribution maps were produced using Surfer 9.0 (Golden Software Inc.) and the Kriging
method of gridding was used for data interpolation.

## 3 Results
### 3.1 Physical characteristics
The sampling sites covered most parts of the Yellow River estuary, with water depth
ranging from 1.5 m to 13.5 m (Figure 2a). Dry bulk density (DBD) ranged from 0.74 to
1.55 g cm$^{-3}$, with an average of 1.02 g cm$^{-3}$ (Table 1). Generally, DBD decreased with
water depth, showing high values mainly occurred in the south and north sides near the
river mouth (Figure 2b).
Figure 3 showed the spatial distributions of the main granulometric variables of the
surface sediment. In general, clay content was low (1.4-10.8%), showing relatively higher
values in the northern part than in the southern part. The highest clay content was found
near the north side of the river mouth, and the lowest at the mouth section. Silt content
was much high (69.4±21.1%), showing similar spatial distribution with clay. On the other
hand, the highest content of sand was found at the mouth (Figure 3c), where clay and silt
contents were lowest (Figure 3a-b). As expected, the spatial distribution of d(0.5) was
similar to that of sand, showing the highest values in the shallow river mouth section and
lowest in the southern bay, indicating strong hydrodynamic effect in the former and weak
in the latter.
### 3.2 Spatial distribution of TOC, TN, C:N and δ$^{13}$C$_{org}$
Concentration of TOC was highly variable, with higher values (3.2-4.4 g kg$^{-1}$) found in the
northernmost section of the estuary and the north side near the mouth (Figure 4a). There
was also high value of TOC in the bay south of the river mouth. On the other hand, lower



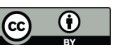

TOC concentration (0.2-1.4 g kg$^{-1}$) was observed near the mouth section. Similarly, TN
value (ranging from 0.06 to 0.68 g kg$^{-1}$) was lowest at the river mouth and highest in the
north section (Figure 4b). Overall, the spatial distribution of TN was similar to that of
TOC, both showing higher values in the north deeper water area.

167        The C:N ratio ranged from 2.1 to 10.1 (Figure 4c). In general, C:N ratio was higher

in the shallow water part relative to the deep water part. The highest C:N ratio (8-10) was
found in the southern bay, and the lowest at the mouth (<4.5). Figure 4d showed a
considerable spatial variability in the $\delta^{13}C_{org}$ values with a range from -24.26‰ to
-22.66‰. The most negative value was observed at the river mouth and its adjacent south
bay, and the least negative value far away from the mouth downward the sea. Overall,
values of $\delta^{13}C_{org}$ were more negative in the shallow water section than in the deep area.

## 3.3    Spatial distribution of TIC, $\delta^{13}C_{carb}$ and $\delta^{18}O_{carb}$

There was a large spatial variation in TIC, as shown in Figure 5a, ranging from 6.3 to 20.1
g kg$^{-1}$, with the highest concentration in the north deep sea area (>16 g kg$^{-1}$) away the
mouth, and the lowest at the river mouth (<10 g kg$^{-1}$). Apparently, TIC also presented a
high value in the southern bay. Overall, the spatial distribution of TIC was similar to that
of TOC. The values of $\delta^{13}C_{carb}$ and $\delta^{18}O_{carb}$ ranged from -4.89‰ to -3.74‰ and -10.92‰
to -7.92‰, respectively (Figure 5b & 5c). Generally, the spatial distribution of $\delta^{13}C_{carb}$ was
opposite to that of $\delta^{18}O_{carb}$, showing more negative values in the north deep sea area.

## 3.4    Relationship between TOC and TIC

As shown in Figure 5, there was a significantly positive correlation between TOC and TIC
in the surface sediments in the Yellow River Estuary (r=0.97, p<0.01). Interestingly, the
slope (2.93) was close to that (2.87) reported for the soils in the upper Yellow River Delta
(Guo et al., 2016). However, the intercept was close to zero for the soils, but 7.17 in the
surface estuarine sediments.





## 4   Discussion

### 4.1   Comparison of TOC with other studies

We first compared TOC concentration in the surface sediment near the river mouth in the Yellow River Estuary. Our value (0.2 to 4.4 g kg$^{-1}$) was slightly lower than the previous reports of 0.7-7.7 g kg$^{-1}$ (Li et al., 2014b) and <1 to 6.0 g kg$^{-1}$ (Liu et al., 2015). Concentration of TOC was lower in our study area relative to those in the other coastal areas of the Bohai Sea, i.e., north off the Yellow River Estuary (2.6-17.2 g kg$^{-1}$) (Yuan et al., 2004) and the Laizhou Bay (5.7-12.8 g kg$^{-1}$) (Wang et al., 2017).

As given in Table 2, the Yellow River Estuary also had relatively lower TOC values than other estuaries in China. There was an increasing trend in TOC from north to south (i.e., Yellow River Estuary < Yangtze River Estuary < Pearl River Estuary), which might be related to the differences in climatic conditions and estuarine sedimentary environments. The warmer and humid climate with a longer growing season to the south would enhance biological production in the water column and sedimentation of organic materials (Dong et al., 2012;Yu et al., 2015a). In addition, the vegetation (i.e., mainly mangroves) grown in the Pearl River Estuary of the South China had much higher carbon sequestration capacity than those (i.e., tidal marshes and seagrass beds) in the Yangtze River Estuary and Yellow River Estuary (Wang et al., 2016d;Pendleton et al., 2012).

Sedimentary TOC concentration in large river estuaries was relatively lower than in small river estuaries, e.g., the Luan River Estuary, Licun Estuary, Min River Estuary and GQ Estuary (Table 2), indicating that the weak hydrodynamic environment (in the small estuaries) was beneficial to the burial of organic carbon (Liu et al., 2015;Ramaswamy et al., 2008). The Yellow River Estuary, and most estuaries in China, generally showed much lower TOC values in the surface sediment than those in the South and Southeast Asia, Europe, North America and South America. The differences in TOC levels may be associated with the geomorphology of the estuary, the magnitude and stoichiometry of nutrient inputs, and other driving mechanisms (Bauer et al., 2013;Cai, 2010).





## 4.2  Dynamics of TOC and regulating mechanisms

Our study demonstrated large spatial variability in the TOC of the surface sediment in the Yellow River Estuary, with relatively higher values in the north section ($2.6 \pm 1.5$ g kg$^{-1}$) than in the south section ($1.6 \pm 0.2$ g kg$^{-1}$) (Table 3). The surface sediments were finer to the north than to the south (Figure 3). In general, fine-grained marine sediments contain higher organic carbon than coarse marine sediments (Canfield, 1994;Hu et al., 2016). On the other hand, coarser (finer) sediment particles usually indicated a stronger (weaker) water energy environment (Molinaroli et al., 2009;Molinaroli et al., 2014). These analyses indicated that the relatively lower TOC values to the south and in the river mouth were attributable to higher kinetic energy level.

The magnitude and spatial distribution of TOC in estuarine sediment may reflect multiple and complex processes (Hu et al., 2016;He et al., 2010). Apart from the estuary's own characteristics, such as the river plume and tidal straining effect (Wang and Wang, 2010;Xu et al., 2013), other factors may have influences on the dynamics of TOC in the surface sediment. For example, land use changes such as industrial and agricultural development would enhance the riverine input of nutrients and organic materials, leading to changes in estuary productivity and TOC burial in the sediment (Yu et al., 2014;Lin et al., 2002;Liu et al., 2012). As shown in Table 4, there was a significantly negative relationship between the $\delta^{13}C_{org}$ value and water depth (r=0.71, p<0.01), implying that the shallow sections in the Yellow River Estuary accumulated more allochthonous OC (with more negative $\delta^{13}C_{org}$ values).

Sedimentary TOC in estuaries may include autochthonous and allochthonous sources (Bianchi and Allison, 2009;Baijulal et al., 2013). Generally, aqueous organic matter has a lower C:N ratio and less negative $\delta^{13}C_{org}$ value than terrigenous source (Lamb et al., 2006;Meyers, 1997). The relatively low C:N ratio ($6.3 \pm 1.7$) and $\delta^{13}C_{org}$ value ($-23.35 \pm 0.48$‰) in our study indicated that TOC was mainly autochthonous in the surface sediment the Yellow River Estuary. However, C:N ratio was relatively higher ($8.8 \pm 1.8$) in the southern shallow bay (Table 3). Such high C:N ratio together with relatively more





negative $\delta^{13}C_{org}$ value (-23.91±0.50‰) (Table 3) suggested that there might be some
allochthonous OC sources. Using the average C:N ratio (10.8 g:g) from the soils collected
near the river mouth (Table 1), and assuming 6.6 mol:mol as the marine end-member, we
estimated that 60.8% of TOC was from soil OC source, indicating that the southern bay
might have accumulated a significant amount of terrigenous OC.

### 4.3   Dynamics of TIC and underlying mechanisms

There have been only a few studies of inorganic carbon from the estuarine sediments
(Table 2). According to these limited studies, the Yellow River Estuary has much higher
TIC values than those (3.3-8.2 g kg$^{-1}$) in the Cochin Estuary, Vellar and Coleroon Estuary,
and Chilika Lagoon of the South Asia. Our study showed large spatial variability in TIC of
the surface sediment in the Yellow River Estuary, with relatively higher values in the north
section (14.5±4.7 g kg$^{-1}$) than in the south section (12.2±1.2 g kg$^{-1}$) (Table 3), which was
consistent with TOC. Our analyses revealed a significantly positive correlation between
TIC and TOC (r=0.97, p<0.01) in the surface sediments, indicating that production of
organic carbon might have a large influence on the formation of carbonate (Paprocka,
2007;Li et al., 2012). Accordingly, it is reasonable to believe that most TIC was from
autogenic carbnate in the surface sediment of the Yellow River Estuary.
The recent study of Liu et al. (2014) demostrated that the rate of CaCO$_3$ precipitation
exceeded that of OC production by a factor of <2 in the water columns of the Yellow
River Estuary whereas the earlier study of Gu et al. (2009) showed a ratio of 3.6 for
IC:OC in particles. We found that TIC concentration was six times of TOC concentration
in the surface sediment of the Yellow River Estuary, which indicated that apart from the
relatively higher CaCO$_3$ production in the upper water column, there would be a decrease
of TOC and/or an increase of TIC during sedimentation and after burial. Organic matter is
often subject to deocmposition process in the surface sediments, which will cause a
decrease in TOC (Alkhatib et al., 2012;Koho et al., 2013;Rieling et al., 2000). Meantime,
CO$_2$ production owing to TOC decomposition would promote carbonate formation in
sediemnts (Zhao et al., 2015;Wang et al., 2016b;Gu et al., 2009). Taking the southern bay





as an example, both TIC and TOC concentrations were relatively higher, and both $\delta^{13}C_{carb}$
and $\delta^{13}C_{org}$ were more negative than in other sections (Table 3). Such relationship
suggested that there might be some transfer of OC to IC in the surface sediment in the bay.
There was a significant negative correlation between $\delta^{13}C_{carb}$ and $\delta^{18}O_{carb}$ in the
Yellow River Estuary (r=-0.84, p<0.01), which was much different to those found in some
inland waters (Yu et al., 2015b;Xu et al., 2006;Wang et al., 2002), which may reflect
complex impacts of various processes in the Yellow River Estuary (e.g., riverian input,
coastal runoff, ocean currents). There have been human activities over the past decades in
the Yellow River Basin, including intensive irrigation and damming, and direct human
regulation of water and sediment discharge (Liu et al., 2014;Gu et al., 2009;Zhang et al.,
2013), which would impact on the biogeochemical processes in the Yellow River Estuary.
Further studies are needed to assess the spatial and temporal variations in the carbon fields
of water column and long sediment core, in order to better understand the carbon cycle in
the Yellow River Estuary and the impacts of human activity and climate change.

**Acknowledgments**
This study is financially supported by the Stat-up fund of Beijing Normal University
(310232102), the China Postdoctoral Science Foundation (2016M600059), the National
Science Foundation of China (41601107), and the Fundamental Research Funds for the
Central Universities.

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



**Table 1.** Means, standard deviation (SD) and coefficients of variation (CV) of the main variables.

| | | DBD | d0.5 | Clay | F-Silt | M-Silt | C-Silt | Sand | TN | TOC | TIC | C/N | $\delta^{13}C_{org}$ | $\delta^{13}C_{PDB}$ | $\delta^{18}O_{PDB}$ |
|---|---|---|---|---|---|---|---|---|---|---|---|---|---|---|---|
| | | g cm⁻³ | μm | % | | | | | g kg⁻¹ | | | | ‰ | | |
| **Sediment** | Mean | 1.02 | 34.2 | 6.1 | 34.8 | 13.7 | 20.9 | 24.5 | 0.36 | 2.3 | 14.1 | 6.3 | -23.35 | -4.36 | -8.92 |
| | SD | 0.20 | 26.3 | 2.9 | 19.0 | 6.7 | 9.4 | 23.8 | 0.19 | 1.3 | 4.0 | 1.7 | 0.48 | 0.41 | 0.90 |
| | CV | 0.20 | 0.77 | 0.47 | 0.55 | 0.49 | 0.45 | 0.97 | 0.53 | 0.57 | 0.28 | 0.27 | -0.02 | -0.09 | -0.10 |
| **Soil** | Mean | / | 28.2 | 4.7 | 31.1 | 25.8 | 25.8 | 12.5 | 0.77 | 8.9 | 12.9 | 10.8 | -22.5 | -4.0 | -9.1 |
| | SD | / | 15.8 | 1.2 | 15.5 | 9.3 | 10.4 | 15.1 | 0.55 | 8.0 | 4.8 | 3.0 | 3.4 | 0.72 | 0.62 |
| | CV | / | 0.56 | 0.26 | 0.50 | 0.36 | 0.40 | 1.21 | 0.71 | 0.90 | 0.37 | 0.27 | -0.15 | -0.18 | -0.07 |

Clay:<2 μm, F-Silt: 2-16 μm, M-Silt:16-32 μm, C-Silt:32-64 μm, Sand:>64 μm



**Table 2.** Summary of TC, TOC and TIC (g kg⁻¹) values in surface sediments from estuaries in the world.

| Region | | Location | TC g kg$^{-1}$ | Mean g kg$^{-1}$ | TOC g kg$^{-1}$ | Mean g kg$^{-1}$ | TIC* g kg$^{-1}$ | Reference |
|---|---|---|---|---|---|---|---|---|
| Aisia | China | Luan River Estuary | | | 0.4-14 | 4.8 | | Li et al. (2016) |
| | | Yellow River Estuary | | | 0.7-7.7 | 4.2 | | Li et al. (2014b) |
| | | Yellow River Estuary | | | <1-6.0 | 3.1 | | Liu et al. (2015) |
| | | Licun Estuary | **6.6-24.5** | **16.4** | **0.2-4.4** | **2.3** | **14.1** | **This study** |
| | | Licun Estuary | | | 6.0-20 | | | Yu et al. (2009) |
| | | Yangtze River Estuary | | | 1.0-7.0 | | | Zhou et al. (2007) |
| | | | | | 1.2-6.8 | | | Li et al. (2014a) |
| | | | | | 0.9-14.3 | | | Liu et al. (2006) |
| | | | | | 0.1-15.9 | 5.2 | | Wang and Xian (2011) |
| | | | | | 6.0-15 | | | Gao et al. (2008) |
| | | Zhangjiang Estuary | | | 7.4-14.9 | | | Xue et al. (2009) |
| | | Min River Estuary | | | 13.6-22.1 | | | Jia et al. (2008) |
| | | Danshui River Estuary | | | 2.9-17.1 | | | Hung et al. (2007) |
| | | Pearl River Estuary | | | 6.0-44.1 | 12.1 | | He et al. (2010) |
| | | | | | 10-14 | | | Ye et al. (2012) |
| | | | | | 0.6-10.2 | 5.4 | | Hu et al. (2006) |
| | | | | | 0.9-28.3 | | | Zhang et al. (2015) |
| | | GQ Estuary | | | 8.5-23.6 | | | Yang et al. (2014) |
| | India | Mandovi Estuary | | | 1.0-30 | 10.5 | | Alagarsamy (1991) |
| | | Mandovi Estuary | | | 1.0-32.3 | | | Nasnolkar et al. (1996) |
| | | Cochin Estuary | 9.8-28.5 | 22.4 | 5.5-25.9 | 19.1 | 3.3 | Gireeshkumar et al. (2013b) |
| | | Cochin Estuary | 8.5-34.1 | 24.3 | 4.2-27.7 | 18.8 | 5.5 | |
| | | Cochin Estuary | 3.0-32 | 20 | 2.6-29.9 | 15.6 | 4.4 | |




| Continent | Country | Estuary | TC range | TC mean | TOC range | TOC mean | TIC | Reference |
|---|---|---|---|---|---|---|---|---|
| | | Kozhikode and Kannur River Estuary | | | 3.0-32.6 | 21 | | Gireeshkumar et al. (2013a) |
| | | | | | 18-70 | | | Manju et al. (2016) |
| | | Ashtamudi River Estuary | | | 9.5-50.2 | 19.4 | | Baijulal et al. (2013) |
| | | Kadinamkulam River Estuary | | | 9.9-77.1 | 31.8 | | |
| | | Godavari Estuary | | | 35-147 | | | Krupadam et al. (2003) |
| | | Vellar Estuary and Coleroon Estuary | 20.6-28.1 | 24.4 | 14.1-18.3 | 16.2 | 8.2 | Prasad and Ramanathan (2008) |
| | Bengal | Chilika lagoon | 3.4-19.7 | 12.1 | 2.6-16.6 | 8.6 | 3.5 | Nazneen and Raju (2017) |
| | Malaysia | Setiu Estuary | | | 7.0-34 | | | Thornberg et al. (2014) |
| | | | | | 1.0-68 | | | Ellis et al. (2014) |
| Europe | Portuguese | Douro River Estuary | | | 0.1-9.3 | | | |
| | | Mondego River Estuary | | | 0-8.2 | | | |
| | | Estremadura River Estuary | | | 0.1-10.7 | | | Dessandier et al. (2016) |
| | | Tagus River Estuary | | | 0.1-7.0 | | | |
| | | Sado River Estuary | | | 3.7-8.4 | | | |
| | France | Loire Estuary | | | 0.2-38.2 | 13.9 | | Coynel et al. (2016) |
| | UK | Humber Estuary | | | 8.6-72.5 | 29.6 | | Andrews et al. (2008) |
| | | Forth Estuary | | | 31-61 | 47 | | Graham et al. (2001) |
| North America | | Gulf of Mexico Microtidal Estuaries | | | 4.6-14 | | | Darrow et al. (2017) |
| | USA | Cedar and Ortega River Estuary | | | 23-226 | 127 | | Ouyang et al. (2006) |
| | | Hudson River Estuary | | | 10.6-27.3 | | | |
| | | Mullica River Estuary | | | 2.2-19.5 | | | Medeiros et al. (2012) |
| | | Pawtuxet River Estuary | | | 0.7-44 | | | Cantwell et al. (2016) |
| South America | Brazil | Caravelas Estuary | | | 0.9-64 | | | Sousa et al. (2016) |

* $TIC = TC_{mean} - TOC_{mean}$






**Table 3.** Means and standard deviation of the carbon variables in different water sections

| Section[#] | | **TOC** | **TIC** | **C/N** | $\delta^{13}C_{org}$ | $\delta^{13}C_{PDB}$ | $\delta^{18}O_{PDB}$ |
|---|---|---|---|---|---|---|---|
| North | Mean | 2.6 | 14.5 | 6.1 | -23.23 | -4.53 | -8.84 |
| | SD | 1.5 | 4.7 | 0.7 | 0.32 | 0.43 | 1.08 |
| South | Mean | 1.6 | 12.2 | 6.2 | -23.11 | -4.03 | -9.1 |
| | SD | 0.2 | 1.2 | 1.1 | 0.26 | 0.18 | 0.42 |
| Bay | Mean | 2.7 | 15.4 | 8.8 | -23.91 | -4.54 | -8.75 |
| | SD | 1.4 | 3.8 | 1.8 | 0.5 | 0.36 | 0.67 |

[#] North: H2, H3, H7, H8, H13, H14; South:H4, H6, H10, H11; Bay: H5, H26

**Table 4.** Correlation coefficient (r) between various variables for the sediments.

| | Depth | DBD | d0.5 | Clay | Silt | Sand | TOC | TIC | $\delta^{13}C_{org}$ | $\delta^{13}C_{carb}$ |
|---|---|---|---|---|---|---|---|---|---|---|
| TOC | 0.54[*] | -0.65[**] | -0.94[**] | 0.97[**] | 0.88[**] | -0.90[**] | | | | |
| TIC | 0.63[*] | -0.70[**] | -0.96[**] | 0.93[**] | 0.93[**] | -0.94[**] | 0.97[**] | | | |
| $\delta^{13}C_{org}$ | 0.71[**] | -0.37 | -0.55[*] | 0.55[*] | 0.53[*] | -0.54[*] | 0.50 | 0.47 | | |
| $\delta^{13}C_{carb}$ | -0.54[*] | 0.73[**] | 0.90[**] | -0.88[**] | -0.85[**] | 0.87[**] | -0.90[**] | -0.93[**] | -0.32 | |
| $\delta^{18}O_{carb}$ | 0.63[*] | -0.65[**] | -0.98[**] | 0.94[**] | 0.96[**] | -0.97[**] | 0.91[**] | 0.93[**] | 0.63[*] | -0.84[**] |

Significance of Pearson correlation is marked with one ($p<0.05$) and two ($p<0.01$) superscripts.



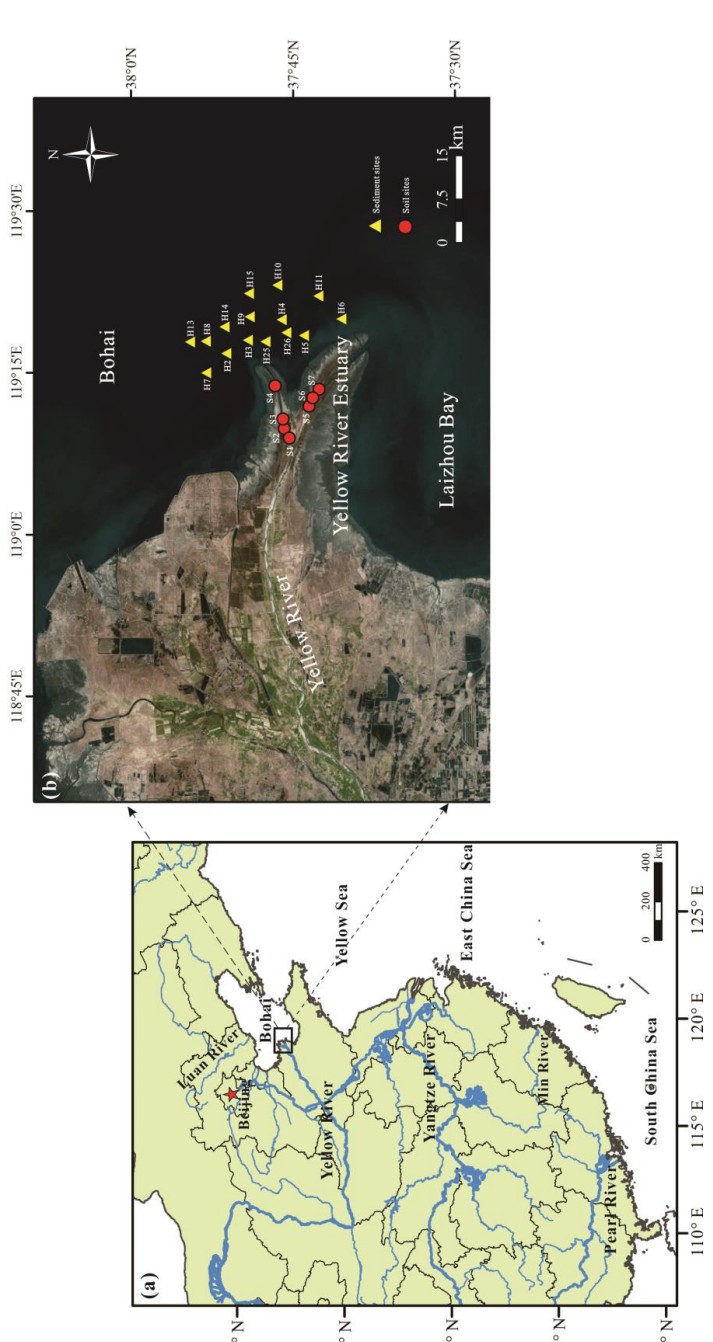

Figure 1. Map of (a) the large Chinese river-estuarine systems and (b) the Yellow River Estuary with the sampling sites. Remote sensing imagery was plotted by using software ArcGIS 10.2 and Corel DRAW X7





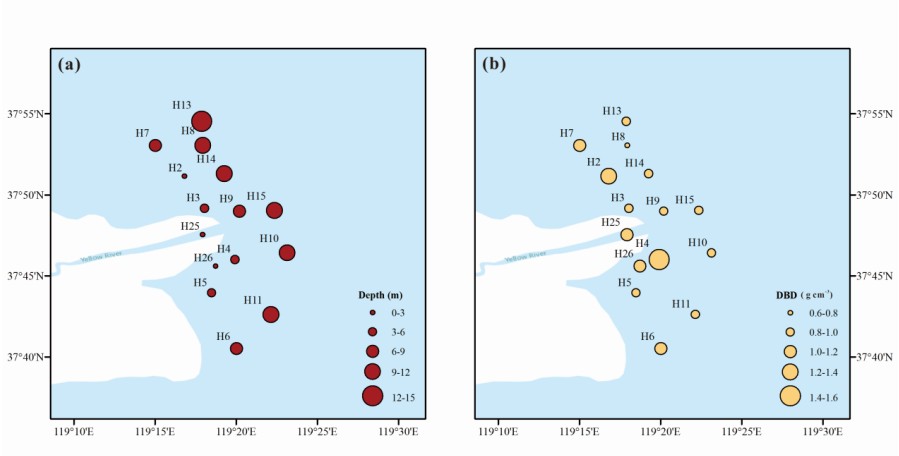

Figure 2. Spatial distributions of **(a)** depth (m) and **(b)** dry bulk density (DBD, g cm$^{-3}$) in surface sediments of the Yellow River Estuary.





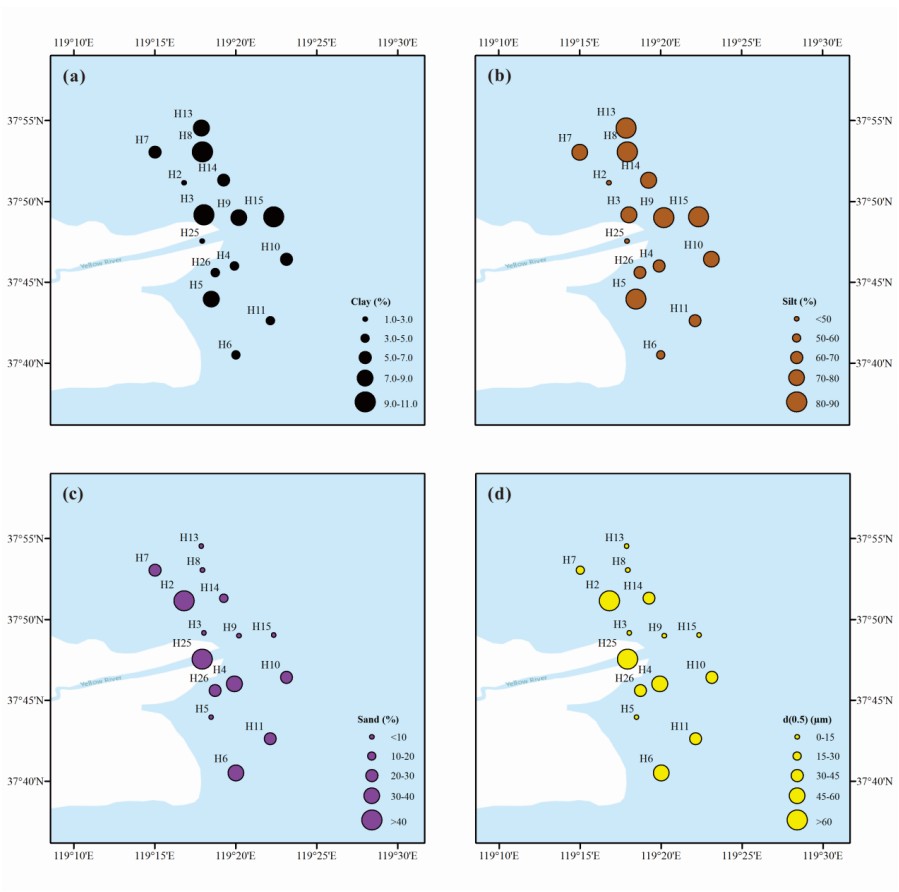

Figure 3. Distributions of **(a)** clay (%), **(b)** silt (%), **(c)** sand (%), **(d)** the median diameter (d(0.5), μm) in surface sediments of the Yellow River Estuary.



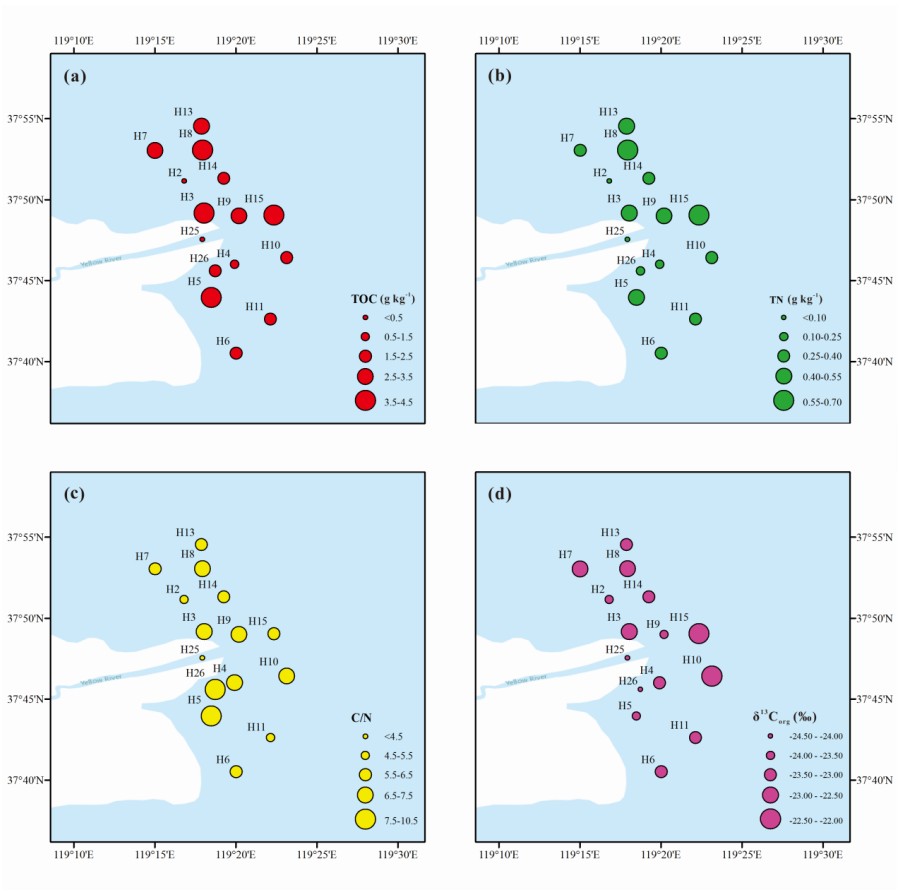

Figure 4. Spatial distributions of **(a)** TOC (g kg$^{-1}$), **(b)** TN (g kg$^{-1}$), **(c)** C/N, **(d)** $\delta^{13}C_{org}$

(‰) in surface sediments of the Yellow River Estuary.



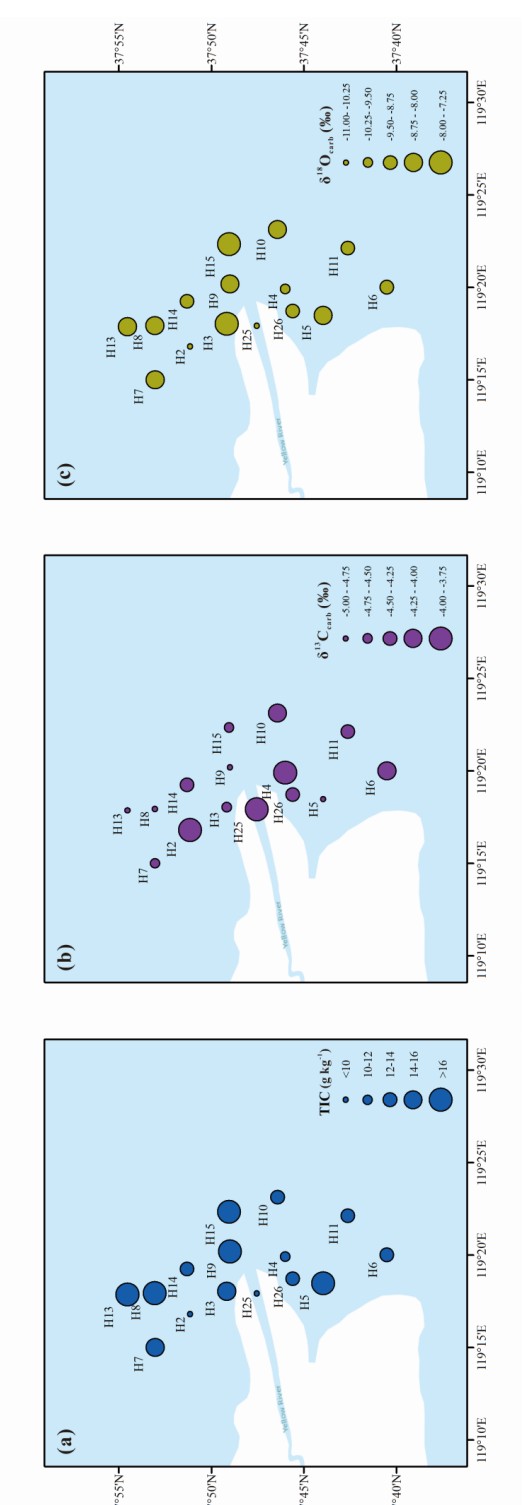

Figure 5. Spatial distributions of **(a)** TIC (g kg$^{-1}$), **(b)** $\delta^{13}C_{carb}$ (‰), and **(c)** $\delta^{18}O_{carb}$ (‰) in surface sediments of the Yellow River Estuary.




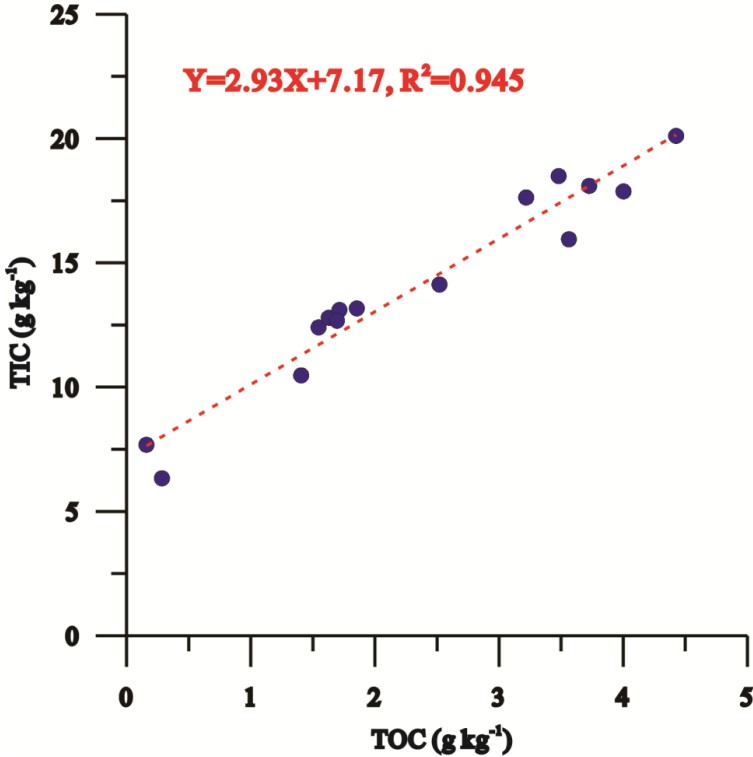

Figure 6. Relationship between TOC and TIC in surface sediments of the Yellow River Estuary.