# Peer review of "Organic and inorganic carbon and their stable isotopes in"

_Biogeosciences, 2017_

## Short Comment (SC1) · 24 Oct 2017

Authors present an important investigation on the magnitudes and spatial distributions of TOC and TIC in the surface sediments of the transitional zone of Yellow River Estuary. Besides, they evaluate the relationship between TOC and TIC, and explore the underlying mechanisms that regulate the carbon burial.

Although its novelty is moderate, the manuscript is clearly written and it contains useful information for potential readers. The discussion of organic and inorganic carbon and their stable isotopes in surface sediments of the Yellow River Estuary are of great importance on the coastal carbon cycle.

[Figure]

I would, however, like to see a more detail discussion of the impact of other sediment/physical characteristics (such as grain size) and human activities on the spatial distributions and transformation of carbon. The end-member modeling for identifying carbon sources is better to be carefully described in the manuscript. At last, a brief summary and/or upscaling are good for this paper and for the journal's international readers.

---

## Referee Comment (RC1) · Anonymous Referee #1 · 8 Dec 2017

The author of this paper present new data of organic and inorganic carbon from the Yellow River Estuary. Authors describe the pattern observed, and conclude that it is a complex system and that some previous findings in the literature are probably right. In my opinion, this is insufficient to be published in Biogeosciences. I do not feel that the conclusions are novel, nor that they are actually based on the new dataset presented here. Find below my evaluation of the paper in regards to the criteria of BG.

1. Does the paper address relevant scientific questions within the scope of BG? No

2. Does the paper present novel concepts, ideas, tools, or data? Data are incremental to Liu et al., 2012, 2014, 2015 (etc.) who all reported similar data from the same region.

3. Are substantial conclusions reached? No, the discussion is very descriptive and the conclusion overly vague: "Our study points out that the dynamics of sedimentary carbon in the Yellow River Estuary is influenced by multiple and complex processes, and highlights the importance of carbonate in carbon sequestration". In my opinion, this is not enough for Biogeosciences. I would expect the author to come up with a precise discussion of the potential processes and at least some hypothesis to test in the future. Furthermore, I would also expect some sort of quantification of the inorganic carbon sequestration, because how can one claim its important if not measured?

4. Are the scientific methods and assumptions valid and clearly outlined? Analytical methods seem fine but assumptions are not clearly outlined and it is hard to understand the logic behind the limited interpretation. Example: "Our analyses revealed a significantly positive correlation between TIC and TOC (r=0.97, p<0.01)". Which statistical test was performed? Is the distribution normal? It doesn't look like it from here. Also, what is the process potentially linking both?

Later in the text, it is stated that when TOC decompose it releases $CO_2$, which promote TIC precipitation. But then, why the relationship is positive and not negative? The relationship should be between TIC and the amount of TOC degraded. Would that be correlated to the total amount of TOC left after degradation? One can raise serious doubt about that. Especially with the relatively small range of concentration. Was any other potential relationship explored? The TOC/TIC and isotopic proxies seem to also follow the same pattern than the composition of the sediment (clay, silt, and sand). Could your distribution simply an effect of different sedimentation regimes?

5. Are the results sufficient to support the interpretations and conclusions? No, I feel the conclusion build more on previous study than the actual data presented here.

6. Is the description of experiments and calculations sufficiently complete and precise to allow their reproduction by fellow scientists (traceability of results)? Yes

7. Do the authors give proper credit to related work and clearly indicate their own

new/original contribution? Yes.

8. Does the title clearly reflect the contents of the paper? Yes

9. Does the abstract provide a concise and complete summary? More or less

10. Is the overall presentation well structured and clear? Yes

11. Is the language fluent and precise? Yes, in general

12. Are mathematical formulae, symbols, abbreviations, and units correctly defined and used? Yes

13. Should any parts of the paper (text, formulae, figures, tables) be clarified, reduced, combined, or eliminated? Yes, the discussion should be improved, to discuss more in depth the different processes in order to come up with a more elaborated conclusion

14. Are the number and quality of references appropriate? Yes

15. Is the amount and quality of supplementary material appropriate? Yes, NA

---

## Referee Comment (RC2) · Anonymous Referee #2 · 19 Dec 2017

In the manuscript BG-2017-353, Yu and coauthors analyzed OC, IC and N contents as well as $\delta$13C and $\delta$15N in 15 short sediments closed to the Yellow River mouth. The river dominated continental margin is the hotspot of global biogeochemical cycle because of the intense interaction between land and sea as well as relatively high primary productivity. Meanwhile, the river dominated margin is characterized by complex environmental processes, making it difficult to quantify the source, transport and burial of organic matter there. So the manuscript addresses an important issue. However, after carefully read it several times, I have to reject it since several data interpretations are misleading. My major concerns are listed as follows.

1) In lower Yellow River and the Bohai Sea, large anthropogenic nutrient inputs caused the eutrophication in the Bohai Sea. For example, Yu et al (2013, Mar. Environ. Sci. 32, 175–177) reported that the total Bohai Sea area with eutrophication status increased from 110 km2 in 1997 to 14080 km2 in 2010. Under this condition, the water and sediments contain significant amount of inorganic nitrogen that inevitably affects the C/N values. Based on method description of the manuscript, the authors did not separate organic and inorganic nitrogen. The C/N as a source indicator is valid only for organic carbon and organic nitrogen (C/N >15 for terrigenous plants and 4-10 for aquatic algae). If they did not pay attention to this point, the estimation of organic matter source based on the C/N is not proper and very likely to underestimate the contribution of terrigenous component, as the authors did in section 4.2 (line 243-248).

2) In the introduction part, the authors claimed that one of their objectives was to explore the underlying mechanisms that regulate the carbon burial in the Yellow River estuary. Unfortunately, I did not see much discussion about this topic. In fact, most of their statements are speculative. For example, from line 256 to 260, given a strong linear correlation between TIC and TOC (r = 0.97, p < 0.01), the authors concluded that the production of organic carbon influences on the formation of carbonate, and most TIC was from autogenic carbon in the Yellow River estuary. This conclusion is very surprising for me. How could it be like this just based on the correlation. A correlation does not mean cause and effect. Furthermore, in the semiarid region of China, such as Loess Plateau, soil contains a lot of inorganic carbon. Since Loess Plateau contributes 90% of the Yellow River's sediment load, the severe soil erosion at the Loess Plateau will bring large amounts of allochthonous organic carbon and inorganic carbon into the Yellow River as well as its estuary. Regarding the degradation of organic matter to produce CO2, the author did not explain at all which mechanism could convert organic matter derived CO2 into carbonate. I don't know either since the extremely turbidity in the Yellow River great restricts the algal growth.

3) In line 241, the authors suggested that TOC was mainly autochthonous in surface

sediments of the Yellow River estuary based on C/N (6.3) and $\delta$13C (-23.35‰, whereas in the southern shallow bay, up to 60.8% of TOC was from soil source give slightly more negative $\delta$13C (-23.91‰ and higher C/N (8.8). Here the author used 10.8 as the terrigenous end member value for C/N based on the soils collected from the river mouth. As I mentioned above, the major sediment load in the Yellow River is not from the soils around the estuary, but from the Loess Plateau in the middle to lower River. Second, there is no much difference in $\delta$13C between the estuary (-23.35‰and southern bay (-23.91‰, so they should have similar organic matter sources. In the northern China marginal seas, C/N ratio is not as reliable as $\delta$13C give the interference of inorganic nitrogen and selected degradation of N-containing organic matter.

Given these drawbacks, I do not think the manuscript meets the criteria of BG.

---

## Short Comment (SC2) · 27 Dec 2017

Estuarine sediment is thought to be hotspots of carbon burial and is therefore important to the global carbon cycle, but unfortunately there are few studies of their carbon cycling and burial in China. This manuscript presented a comprehensive analysis of organic carbon and inorganic carbon and other geochemical data from 15 short sediment cores taken from the Yellow River Estuary. The authors studied the spatial variability and their influencing factors of TOC and TIC concentrations in the estuary and concluded that t the dynamics of sedimentary carbon in the Yellow River Estuary is influenced by multiple and complex processes. This study provides a useful budget estimate

of carbon burial from this estuary, which could be useful for future regional or global syntheses of carbon cycling.
* * *

---

## Author Comment (AC1) · 10 Feb 2018

We appreciate the editor's precious time for handling our manuscript and the reviewers' time for reviewing the manuscript. We have thoroughly considered all the comments that are very helpful for improving the interpretations of our findings. We provide our detailed responses below.

Response to the Referees Anonymous Referee #1 1. Does the paper address relevant scientific questions within the scope of BG? No.

Response: Biogeosciences (BG) is an international scientific journal dedicated to the

publication and discussion on all aspects of the interactions between the biological, chemical, and physical processes in terrestrial or extraterrestrial life with the geo-sphere, hydrosphere, and atmosphere. Our study is designed to investigate the spatial distributions of TOC and TIC in the surface sediment of the transitional zone near the Yellow River's mouth, which is influenced by complex interactions of biological, chem-ical, and physical processes. Our analyses address the underlying mechanisms that regulate the carbon sedimentation in the Yellow River Estuary. In this regard, we be-lieve that our paper address relevant scientific questions within the scope of BG.

2. Does the paper present novel concepts, ideas, tools, or data? Data are incremental to Liu et al., 2012, 2014, 2015 (etc.) who all reported similar data from the same region.

Response: There were only a few studies carried out to evaluate the relevant carbon parameters (i.e., DIC, PIC, DOC, POC, TOC) in sections close to or including some areas of the Yellow River Estuary (see Figure 1). Our study differs largely from the previous studies in terms of both sampling area and analyzed variables. Regarding the variables, Liu et al. (2015) and Hu et al. (2016) focused on TOC in sediment, and Gu et al. (2009) and Liu et al. (2014) mainly on DIC, DOC, PIC and/or POC in the water column. However, our study included the analyses of both TOC and TIC in surface sediment of the Yellow River Estuary, which has been lacking although Liu et al. (2014) and Gu et al. (2009)'s analyses pointed out the importance of $CaCO_3$ precipitation in the estuary. Figure 1. Sampling locations and measured variables from previous studies and our study

3. Are substantial conclusions reached? No, the discussion is very descriptive and the conclusion overly vague: "Our study points out that the dynamics of sedimentary carbon in the Yellow River Estuary is influenced by multiple and complex processes, and highlights the importance of carbonate in carbon sequestration". In my opinion, this is not enough for Biogeosciences. I would expect the author to come up with a precise discussion of the potential processes and at least some hypothesis to test in the future. Furthermore, I would also expect some sort of quantification of the inorganic

carbon sequestration, because how can one claim its important if not measured?

Response: We appreciate the reviewer's constructive comment. Our analysis shows a significantly negative relationship between $\delta$13Ccarb and TIC, indicating that higher level of TIC is a result of higher rate of biological production, which would lead to more negative $\delta$13Ccarb. Thus, TIC in the surface sediment of Yellow River Estuary is primarily from autogenic carbonate. Interestingly, there is also a significantly negative relationship between $\delta$13Ccarb and TOC, implying that higher level of TOC may also result from higher rate of biological production, thus TOC is primarily autochthonous. However, we agree that the discussion and interpretation need to be improved. Author's changes in manuscript: We will revise our discussion/interpretation, and also make changes in other relevant sections. For example, in Abstract, we will include statements similar to "there is a significantly negative relationship between TIC and $\delta$13Ccarb, indicating that TIC was primarily from autogenic carbonate", and "our analysis shows a significantly negative correlation between $\delta$13Ccarb and TOC, implying that TOC is mainly autochthonous".

4. Are the scientific methods and assumptions valid and clearly outlined? Analytical methods seem fine but assumptions are not clearly outlined and it is hard to understand the logic behind the limited interpretation. Example: "Our analyses revealed a significantly positive correlation between TIC and TOC (r=0.97, p<0.01)". Which statistical test was performed? Is the distribution normal? It doesn't look like it from here. Also, what is the process potentially linking both?

Response: We appreciate the reviewer's constructive comment. In our study, a correlation analysis was carried out to evaluate the relationship between TIC and TOC. Student's t test was used to determine the correlation's significance. The distribution is normal. Regarding "the process potentially linking both", our response to the second comment is relevant, i.e., higher levels of TOC and TIC are primarily a result of higher rate of photosynthesis. However, we agree that the discussion/interpretation need improvements. Author's changes in manuscript: We will add the relevant information and

provide in-depth analyses with "precise discussion of the potential processes" in our revision.

5. Later in the text, it is stated that when TOC decompose it releases CO2, which promote TIC precipitation. But then, why the relationship is positive and not negative? The relationship should be between TIC and the amount of TOC degraded. Would that be correlated to the total amount of TOC left after degradation? One can raise serious doubt about that. Especially with the relatively small range of concentration. Was any other potential relationship explored? The TOC/TIC and isotopic proxies seem to also follow the same pattern than the composition of the sediment (clay, silt, and sand).Could your distribution simply an effect of different sedimentation regimes?

Response: This is a good point. The statement/interpretation (i.e., OC transfer to IC) we gave earlier is not appropriate. We have re-evaluated our analyses and interpretations, and intend to revise our discussion and interpretation regarding the underlying mechanisms responsible for the spatial distributions of TOC and TIC (see responses to comments 2 and 3). Author's changes in manuscript: We will make changes accordingly during the revision.

6. Are the results sufficient to support the interpretations and conclusions? No, I feel the conclusion build more on previous study than the actual data presented here.

Response: As given in our responses above, while our discussion and interpretation need some improvements, the main conclusions are almost correct. We believe that with a modest to major revision, our results will be sufficient to support the interpretations and conclusions. Although previous studies have pointed out the importance of TIC near the Yellow River Estuary, there was no measurement to support it. Our study is the first to evaluate both TOC and TIC in the surface sediment, and to explore the underlying processes determining the dynamics of TOC and TIC. Author's changes in manuscript: We will discuss more in depth the different processes in order to come up with more elaborated interpretations and conclusions in the revision.

Gu, D., Zhang, L., and Jiang, L.: The effects of estuarine processes on the fluxes of inorganic and organic carbon in the Yellow River estuary, Journal of Ocean University of China, 8, 352-358, 10.1007/s11802-009-0352-x, 2009. Hu, L., Shi, X., Bai, Y., Qiao, S., Li, L., Yu, Y., Yang, G., Ma, D., and Guo, Z.: Recent organic carbon sequestration in the shelf sediments of the Bohai Sea and Yellow Sea, China, Journal of Marine Systems, 155, 50-58, https://doi.org/10.1016/j.jmarsys.2015.10.018, 2016. Liu, D., Li, X., Emeis, K.-C., Wang, Y., and Richard, P.: Distribution and sources of organic matter in surface sediments of Bohai Sea near the Yellow River Estuary, China, Estuarine, Coastal and Shelf Science, 165, 128-136, https://doi.org/10.1016/j.ecss.2015.09.007, 2015. Liu, Z., Zhang, L., Cai, W.-J., Wang, L., Xue, M., and Zhang, X.: Removal of dissolved inorganic carbon in the Yellow River Estuary, Limnology and Oceanography, 59, 413-426, 2014.

Please also note the supplement to this comment:
https://www.biogeosciences-discuss.net/bg-2017-353/bg-2017-353-AC1-supplement.pdf
* * *
**Fig. 1.** Sampling locations and measured variables from previous studies and our study

---

## Author Comment (AC2) · 10 Feb 2018

We appreciate the editor's precious time for handling our manuscript and the reviewers' time for reviewing the manuscript. We have thoroughly considered all the comments that are very helpful for improving the interpretations of our findings. We provide our detailed responses below.

Response to the Referees Anonymous Referee #2

1. In lower Yellow River and the Bohai Sea, large anthropogenic nutrient inputs caused the eutrophication in the Bohai Sea. For example, Yu et al (2013, Mar. Environ. Sci. 32,

175–177) reported that the total Bohai Sea area with eutrophication status increased from 110 km2 in 1997 to 14080 km2 in 2010. Under this condition, the water and sediments contain significant amount of inorganic nitrogen that inevitably affects the C/N values. Based on method description of the manuscript, the authors did not separate organic and inorganic nitrogen. The C/N as a source indicator is valid only for organic carbon and organic nitrogen (C/N >15 for terrigenous plants and 4-10 for aquatic algae). If they did not pay attention to this point, the estimation of organic matter source based on the C/N is not proper and very likely to underestimate the contribution of terrigenous component, as the authors did in section 4.2 (line 243-248).

Response: While Yu et al., (2013) reported an increasing trend in eutrophication area from 1997 to 2010 in the Bohai Sea, the most heavily eutrophic water was mainly in these bays (Wang et al., 2015). According to the recent studies (Liu et al., 2015;Zhang et al., 2012), $\delta$15N values in the Yellow River mouth were 4-5‰ which were much different from the elevated $\delta$15N values (10-25‰ delivered from farm runoff and human sewage. These findings indicate that inorganic nitrogen should not be a concern. On the other hand, the approach using the TOC/TN ratio is a common method to quantify different sources of OC, which has been widely applied to study wetland and lake sediments (Meyers, 1997;Brodie et al., 2011;Kaushal and Binford, 1999), and offshore and marine sediments (Lamb et al., 2006;Rumolo et al., 2011). Nevertheless, we will conduct an uncertainty analysis to assess the potential underestimation of terrigenous component. Author's changes in manuscript: In our revision, we will add a section of uncertainty analysis by assuming different degrees of underestimation for the C/N ratio.

2. In the introduction part, the authors claimed that one of their objectives was to explore the underlying mechanisms that regulate the carbon burial in the Yellow River estuary. Unfortunately, I did not see much discussion about this topic. In fact, most of their statements are speculative. For example, from line 256 to 260, given a strong linear correlation between TIC and TOC (r = 0.97, p < 0.01), the authors concluded that the production of organic carbon influences on the formation of carbonate, and most

TIC was from autogenic carbon in the Yellow River estuary. This conclusion is very surprising for me. How could it be like this just based on the correlation. A correlation does not mean cause and effect.

Response: We appreciate the reviewer's constructive comment. We have re-evaluated our analyses and interpretations, and intend to revise our discussion and interpretation regarding the underlying mechanisms responsible for the spatial distributions of TOC and TIC. Author's changes in manuscript: Our analysis shows a significantly negative relationship between $\delta$13Ccarb and TIC, indicating that higher level of TIC is a result of higher rate of biological production, which would lead to more negative $\delta$13Ccarb. Thus, TIC in the surface sediment of Yellow River Estuary is primarily from autogenic carbonate. Interestingly, there is also a significantly negative relationship between $\delta$13Ccarb and TOC, implying that higher level of TOC may also result from higher rate of biological production, thus TOC is primarily autochthonous.

3. Furthermore, in the semiarid region of China, such as Loess Plateau, soil contains a lot of inorganic carbon. Since Loess Plateau contributes 90% of the Yellow River's sediment load, the severe soil erosion at the Loess Plateau will bring large amounts of allochthonous organic carbon and inorganic carbon into the Yellow River as well as its estuary. Regarding the degradation of organic matter to produce CO2, the author did not explain at all which mechanism could convert organic matter derived CO2 into carbonate. I don't know either since the extremely turbidity in the Yellow River great restricts the algal growth.

Response: According to Wang et al. (2016), the Yellow River sediment load has decreased since the 1950s (see Figure 1 below). Thus, it is reasonable to assume that Loess's contribution is small. Our re-analyses lead to the following interpretations and conclusions: "Our analysis shows a significantly negative relationship between $\delta$13Ccarb and TIC, indicating that higher level of TIC is a result of biological production, which would lead to more negative $\delta$13Ccarb. Thus, TIC in the surface sediment of Yellow River Estuary is primarily from autogenic carbonate. Interestingly,

there is also a significantly negative relationship between $\delta13Ccarb$ and TOC, implying that higher level of TOC may also result from higher rate of biological production, thus TOC is primarily autochthonous." Figure 1 The hydrologic regime of the Yellow River in recent decades (Wang et al., 2016)

Author's changes in manuscript: We will revise our discussion/interpretation /conclusion, and also make changes in other relevant sections (e.g., Abstract).

4. In line 241, the authors suggested that TOC was mainly autochthonous in surface sediments of the Yellow River estuary based on C/N (6.3) and $\delta13C$ (-23.35‰ whereas in the southern shallow bay, up to 60.8% of TOC was from soil source give slightly more negative $\delta13C$ (-23.91‰ and higher C/N (8.8). Here the author used 10.8 as the terrigenous end member value for C/N based on the soils collected from the river mouth. As I mentioned above, the major sediment load in the Yellow River is not from the soils around the estuary, but from the Loess Plateau in the middle to lower River. Second, there is no much difference in $\delta13C$ between the estuary (-23.35‰ and southern bay (-23.91‰ so they should have similar organic matter sources. In the northern China marginal seas, C/N ratio is not as reliable as $\delta13C$ give the interference of inorganic nitrogen and selected degradation of N-containing organic matter.

Response: We appreciate the reviewer's constructive comment. In our earlier responses, "inorganic nitrogen should not be a concern", and "Loess's contribution is small". Our re-evaluation based on the significantly negative relationship between $\delta13Ccarb$ and TOC suggests that TOC is primarily autochthonous. However, our approach using the two-end-member mixing model may introduce bias or uncertainty due to the choice of end member value for soil C/N ratio. For example, our recent study shows a wide range of soil C/N ratio in the lower Yellow River Basin, with a mean value of ~10 (Shi et al., 2017). We will carry out an uncertainty analysis to address this issue. Author's changes in manuscript: We will revise the discussion/interpretation and add a section uncertainty analysis.

Brodie, C. R., Leng, M. J., Casford, J. S. L., Kendrick, C. P., Lloyd, J. M., Yongqiang, Z., and Bird, M. I.: Evidence for bias in C and N concentrations and $\delta$13C composition of terrestrial and aquatic organic materials due to pre-analysis acid preparation methods, Chemical Geology, 282, 67-83, http://dx.doi.org/10.1016/j.chemgeo.2011.01.007, 2011. Kaushal, S., and Binford, M.: Relationship between C:N ratios of lake sediments, organic matter sources, and historical deforestation in Lake Pleasant, Massachusetts, USA, J Paleolimnol, 22, 439-442, 1999. Lamb, A. L., Wilson, G. P., and Leng, M. J.: A review of coastal palaeoclimate and relative sea-level reconstructions using $\delta$13C and C/N ratios in organic material, Earth-Science Reviews, 75, 29-57, http://dx.doi.org/10.1016/j.earscirev.2005.10.003, 2006. Liu, D., Li, X., Emeis, K.-C., Wang, Y., and Richard, P.: Distribution and sources of organic matter in surface sediments of Bohai Sea near the Yellow River Estuary, China, Estuarine, Coastal and Shelf Science, 165, 128-136, https://doi.org/10.1016/j.ecss.2015.09.007, 2015. Meyers, P. A.: Organic geochemical proxies of paleoceanographic, paleolimnologic, and paleoclimatic processes, Organic Geochemistry, 27, 213-250, http://dx.doi.org/10.1016/S0146-6380(97)00049-1, 1997. Rumolo, P., Barra, M., Gherardi, S., Marsella, E., and Sprovieri, M.: Stable isotopes and C/N ratios in marine sediments as a tool for discriminating anthropogenic impact, Journal of Environmental Monitoring, 13, 3399-3408, 2011. Wang, R., Tang, J., Huang, G., Chen, Y., Tian, C., Pan, X., Luo, Y., Li, J., and Zhang, G.: Provenance of organic matter in estuarine and marine surface sediments around the Bohai Sea, Oceanologia et Limnologia Sinica, 46, 497-507, 2015. Wang, S., Fu, B., Piao, S., Lü, Y., Ciais, P., Feng, X., and Wang, Y.: Reduced sediment transport in the Yellow River due to anthropogenic changes, Nature Geoscience, 9, 38-41, 2016. Zhang, D., Yang, W., and Zhao, J.: Tracing nitrate sources of the Yellow River and its tributaries with nitrogen isotope, Journal of Ecology & Rural Environment, 28, 622-627, 2012.

Please also note the supplement to this comment:
https://www.biogeosciences-discuss.net/bg-2017-353/bg-2017-353-AC2-

[Figure]

supplement.pdf

[Figure]

**Fig. 1.** The hydrologic regime of the Yellow River in recent decades

---

## Author Comment (AC3) · 10 Feb 2018

We appreciate Dr. Zhang's constructive comments. Many thanks!